

# Results from recent analysis of KASCADE-Grande data

D. Kang[1][*], J.C. Arteaga-Velázquez[2], M. Bertaina[3], A. Chiavassa[3], K. Daumiller[1],
V. de Souza[4], R. Engel[1,5], A. Gherghel-Lascu[6], C. Grupen[7], A. Haungs[1],
J.R. Hörandel[8], T. Huege[1], K.-H. Kampert[9], K. Link[1], H.J. Mathes[1], S. Ostapchenko[10],
T. Pierog[1], D. Rivera-Rangel[2], M. Roth[1], H. Schieler[1], F.G. Schröder[1], O. Sima[11],
A. Weindl[1], J. Wochele[1] and J. Zabierowski[12]

1 Karlsruhe Institute of Technology, Institute for Astroparticle Physics, Germany
2 Universidad Michoacana, Instituto de Física y Matemáticas, Morelia, Mexico
3 Dipartimento di Fisica, Università degli Studi di Torino, Italy
4 Universidade São Paulo, Instituto de Física de São Carlos, Brasil
5 Karlsruhe Institute of Technology, Institute of Experimental Particle Physics, Germany
6 Horia Hulubei National Institute of Physics and Nuclear Engineering, Bucharest, Romania
7 Department of Physics, Siegen University, Germany
8 Dept. of Astrophysics, Radboud University Nijmegen, The Netherlands
9 Fachbereich Physik, Universität Wuppertal, Germany
10 Hamburg University, II Institute for Theoretical Physics, 22761 Hamburg
11 Department of Physics, University of Bucharest, Bucharest, Romania
12 National Centre for Nuclear Research, Department of Astrophysics, Lodz, Poland

★ donghwa.kang@kit.edu

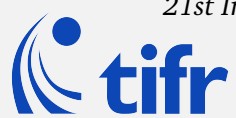

*21st International Symposium on Very High Energy Cosmic Ray Interactions
(ISVHECRI 2022)
Online, 23-28 May 2022*

## Abstract

**KASCADE and its extension array of KASCADE-Grande were devoted to measure individual air showers of cosmic rays in the primary energy range of 100 TeV to 1 EeV. The experiment has substantially contributed to investigate the energy spectrum and mass composition of cosmic rays in the transition region from galactic to extragalactic origin of cosmic rays as well as to quantify the characteristics of hadronic interaction models in the air shower development through validity tests using the multi-detector information from KASCADE-Grande. Although the data accumulation was completed in 2013, data analysis is still continuing. Recently, we investigated the reliability of the new hadronic interactions models of the SIBYLL version 2.3d only with the energy spectra from the KASCADE-Grande data. The evolution of the muon content of high energy air showers in the atmosphere is studied as well, using EPOS-LHC, SIBYLL 2.3, QGSJET-II-04 and SIBYLL 2.3c. In this talk, recent results from KASCADE-Grande and the update of the KASCADE Cosmic Ray Data Centre (KCDC) will be discussed.**

# 1 Introduction

Measurements of high-energy cosmic rays are dedicated to understand the chemical mass composition, the energy spectrum and the arrival direction of cosmic rays. These investigations give an important clue on the origin of cosmic rays, and their acceleration and propagation mechanisms. In particular, KASCADE and KASCADE-Grande contributed to identifying the transition region of galactic and extragalactic cosmic rays.

The KASCADE [1] and its extension KASCADE-Grande [2] experiments were located at the Karlsruhe Institute of Technology, Karlsruhe, Germany (49.1° north, 8.4° east, 110 m above sea level) and measured extensive air showers in the energy range of PeV to EeV. One remark is that the multi-detector setup of KASCADE allowed us to reconstruct the number of electrons and muons for individual air showers separately.

The operation for the data accumulation was completed at the end of 2013 and all detector components are fully dismantled. Detailed analysis of more than 20 years of measured data presents fruitful results. In terms of the KASCADE measurement, the reconstructed all-particle energy spectrum shows a knee-like structure, mainly, due to a steepening of spectra of light (H and He) elements [3]. The all-particle energy spectrum measured by KASCADE-Grande [4] shows a concave behavior just above $10^{16}$ eV and a small break at around $10^{17}$ eV, where a knee-like feature would be expected as the knee of the heavy primaries, mainly, iron. The knee-like structure in the spectrum of heavy mass group is observed more significantly at around 80 PeV [5]. Furthermore, an ankle-like structure is observed at an energy of 100 PeV in the energy spectrum of light primary cosmic rays [6].

The data collection was completed, though the analysis of the measured data in more than 20 years is still in progress. In particular, we use the data to investigate the validity of the new hadronic interaction models. In addition, the KASCADE and KASCADE-Grande experiments were able to measure the number of muons with high purity. Measurements of KASCADE-Grande on the muon number suggest that the attenuation length of muons in the atmosphere is larger than the predictions from the hadronic interaction models. In order to investigate the muon anomalies, we also estimated the number of muons as a function of the primary energy at different zenith angles using the data from KASCADE-Grande.

In this paper, we will discuss on recently ongoing studies: the testing of the new interaction model of SIBYLL 2.3d and the study of the muon content. Finally, the KASCADE Cosmic ray Data Center (KCDC) will be briefly discussed as well.

# 2 Testing hadronic interaction model

## 2.1 SIBYLL 2.3d

One of the most important analysis after finishing data taking of KASCADE and KASCADE-Grande is the validity test of new hadronic interaction models. Recently, a new version of SIBYLL, SIBYLL 2.3d, was released [7]. The updated model is developed by improving the behavior of the $\pi^{\pm}$ to $\pi^{0}$ ratio in different mechanisms of hadronization, in order to improve the description of extensive air showers, in particular, the muon content, which impacts the interpretation of the mass composition of the primary cosmic rays. Regarding the impact on extensive air showers, the muon number in the updated model increased by more than 20% compared to SIBYLL 2.1. While SIBYLL 2.1 predicted lower muon numbers, those in SIBYLL 2.3d are about 5% higher as compared to EPOS-LHC [8] and QGSJET-II-04 [9]. Further details can be found in Ref. [7].

The program CORSIKA [10] has been used for the air shower simulations, applying different embedded hadronic interaction models. The FLUKA (E < 200 GeV) model has been used for hadronic interactions at low energies, while high-energy interactions were treated with SIBYLL 2.3d. Air showers induced by five different primaries (H, He, CNO, Si, and Fe nuclei) have been simulated. The simulations covered the energy range of $10^{14}$ - $10^{18}$ eV with zenith angles in the interval 0°- 42°. The spectral index in the simulations was -2 and for the analysis it is weighted to a slope of -3.

## 2.2 Spectra of heavy and light mass groups

Based on the measured shower size by KASCADE-Grande only, we reconstructed the primary energy spectrum [11] using the energy calibration with the new SIBYLL 2.3d model, where the atmospheric attenuation effects are corrected by using the Constant Intensity Cut (CIC) method.

To reconstruct energy spectra for individual mass groups, we divided the data into two subsets for heavy and light groups, based on the $y_{CIC}$ cut method [4]. It is the shower size ratio of the attenuation-corrected muon and charged particle numbers: $y_{CIC} = \log_{10}(N_{\mu})$ / $\log_{10}(N_{ch})$. The events satisfying the condition ($y_{CIC} \geq y_{CIC}^{thr}$) are defined as electron-poor (heavy) events and the remaining as electron-rich (light) events. The value of the selection criteria of $y_{CIC}^{thr}$ depends on the interaction models and it is defined to be between the CNO and the silicon elements for each models.

Figure 1 presents the energy calibration function for light and heavy induced showers. With the assumption of a linear dependence in logarithmic scale ($\log_{10}E = a \cdot \log_{10}(N_{ch}) + b$), the fitting is applied in the range of full trigger and reconstruction efficiencies. The resulting coefficients of the energy calibration for SIBYLL 2.3d are $a = 0.891 \pm 0.004$, $b = 1.802 \pm 0.0244$ and $a = 0.943 \pm 0.005$, $b = 1.216 \pm 0.035$ for heavy and light primaries, respectively.

The energy spectra of heavy and light mass groups are reconstructed using the relation $E(N_{ch})$ for two separated samples using the energy calibration function, which has model dependency, i.e. we converted the attenuation corrected shower size into the reconstructed energy. Figure 2 shows the reconstructed energy spectra for light and heavy initiated showers with only statistical errors. Systematic uncertainties are expected to be about 25% on the flux, however, a detailed estimation is still in progress.

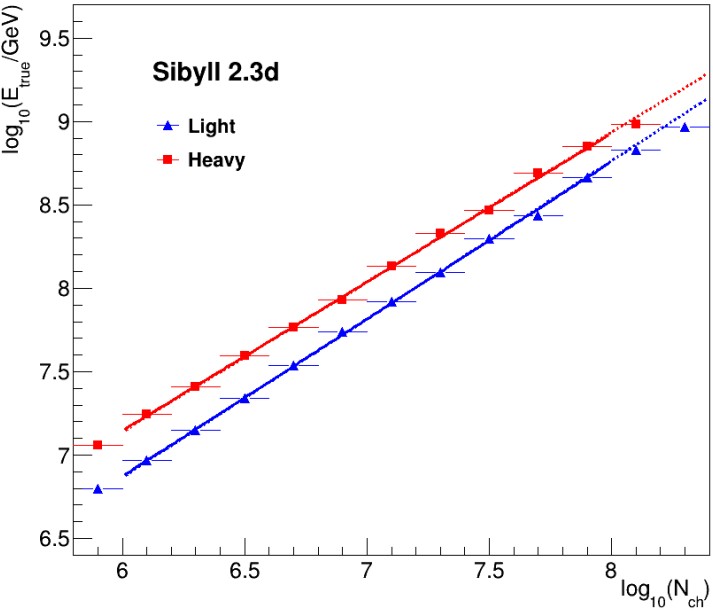

Figure 1: The true primary energy as a function of the number of charged particles ($N_{ch}$) for light (blue) and heavy (red) primaries for SIBYLL 2.3d.

A broken power-law fitting has been performed for the spectra (Fig. 2). Table 1 summarizes the resulting slopes before and after the heavy knee and the positions of the spectral breaks. All features observed by the previous analysis are well confirmed. The spectrum of heavy primaries, i.e. electron-poor events, shows a clear knee-like structure at $10^{16.7}$ eV. A remarkable feature is that the concave structure at about $10^{16}$ eV is more visible in the spectrum of electron-poor components. A hardening of the spectrum is expected when a pure rigidity dependence of the galactic cosmic rays is assumed. The gap in the knee positions of light primaries and the heavy group can lead to a hardening of the spectrum. A detailed discussion on this structure can be found in Ref. [5]. In the energy spectrum of the light primaries, a hardening feature above about $10^{16.5}$ eV is significantly observed and the spectral slope changes smoothly. It is interesting to note that the spectrum for light primaries gets very close to the one for heavy primary at energies around 1 EeV.

The comparison of the reconstructed energy spectra of the electron-poor and electron-rich mass groups to other post-LHC models of QGSJET-II-04, EPOS-LHC, SIBYLL 2.3 and SIBYLL 2.3d, and the pre-LHC model SIBYLL 2.1 is shown in Fig. 3., where all spectra were reconstructed by applying the CIC technique. In comparison to the previous model SIBYLL 2.1, we see differences by a factor of about 3 in the flux multiplied by $E^{2.7}$ of heavy primaries w.r.t. SIBYLL 2.3d. The total flux is shifted by about 10-20% due to the different ratio of $N_{ch}/N_\mu$, however, all the spectra show a similar feature of the energy spectrum. The muon content might affect the difference of absolute abundances and further details are discussed in the next section.

## 3 Muon content

The muon component of extensive air showers is a sensitive observable for the mass composition of primary cosmic rays and the physics of hadronic interaction. Recently, the mea-

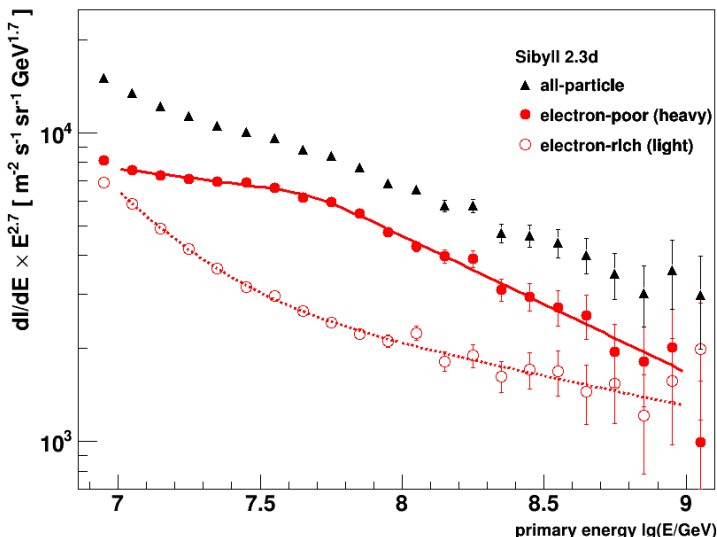

Figure 2: The resulting energy spectra of heavy (solid circles) and light (open circles) primaries based on the SIBYLL 2.3d model fitting with a broken power law.

Table 1: The positions of the spectral breaks and the spectral slopes after applying a broken power law fit to the spectra of electron-poor events.

| electron-poor | $\log_{10}(E_k/\text{GeV})$ | $\gamma_1$ | $\gamma_2$ | $\Delta\gamma$ | $\chi^2/\text{ndf}$ |
|---|---|---|---|---|---|
| QGSjet-II-04 | $7.73 \pm 0.05$ | $2.89 \pm 0.01$ | $3.18 \pm 0.04$ | 0.29 | 2.16 |
| EPOS-LHC | $7.79 \pm 0.03$ | $2.87 \pm 0.01$ | $3.20 \pm 0.03$ | 0.33 | 4.72 |
| SIBYLL 2.1 | $7.75 \pm 0.09$ | $2.87 \pm 0.03$ | $3.15 \pm 0.05$ | 0.28 | 1.28 |
| SIBYLL 2.3 | $7.71 \pm 0.05$ | $2.83 \pm 0.01$ | $3.18 \pm 0.05$ | 0.35 | 0.96 |
| SIBYLL 2.3d | $7.69 \pm 0.04$ | $2.82 \pm 0.01$ | $3.14 \pm 0.03$ | 0.32 | 1.47 |

surements of the muon content in extensive air showers of several experiments exhibited discrepancies between the data and the predictions of high-energy interaction models [16]. In particular, the analysis results show an excess of the measured number of shower muons over predictions, which increases with the primary energy. In addition, the all-particle energy spectra for the different interaction models show a similar structure, though they still do not agree with each other and do also not agree with data. This problem might be caused by the muons.

## 3.1 Muon attenuation length

For this purpose, we performed the analysis of the attenuation length of the number of shower muons in the atmosphere, $\Lambda_\mu$, as an effective physical quantity to study the evolution of the muon content of extensive air showers. This is a rather direct way to compare the $N_\mu$ evolution observed in extensive air showers with the predictions from Monte Carlo simulations. The muon attenuation length is obtained by means of a global fit to the attenuation curves of shower muons using

$$N_\mu(\theta) = N_\mu^0 e^{-X_0 \sec(\theta)/\Lambda_\mu} , \tag{1}$$

where $X_0 = 1022$ g/cm$^2$ is the average atmospheric depth for vertical showers at the location of the KASCADE experiment and $N_\mu$ is a normalization parameter to be determined for each attenuation curve. These curves were obtained using the Constant Intensity Cut method on

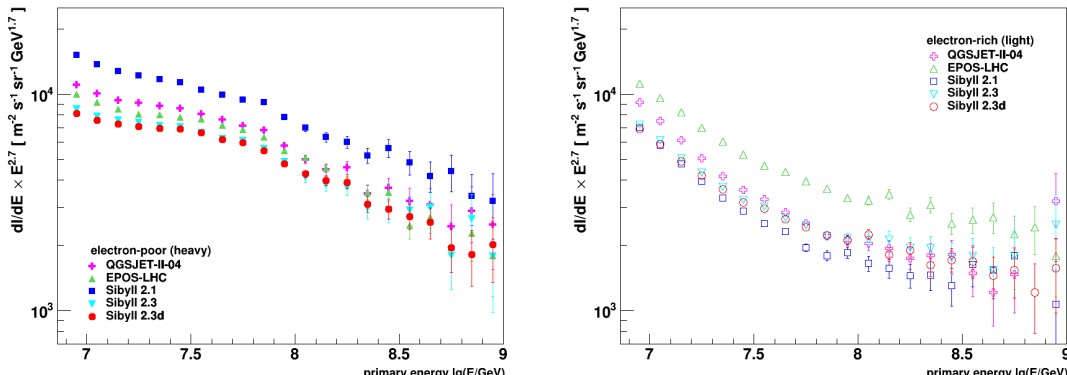

Figure 3: Comparisons of the reconstructed energy spectra of heavy (left) and light (right) mass groups for five hadronic interaction models. The error bars show the statistical uncertainties.

the muon integral fluxes of different zenith angle intervals. More detailed description of the analysis technique can be found in Ref. [12].

In KASCADE-Grande, we obtained a muon attenuation length in the atmosphere of $\Lambda_\mu = 1256 \pm 85(\text{stat})$ g/cm$^2$ from the experimental data for shower energies between $10^{16.3}$ and $10^{17}$ eV [12]. Figure 4 displays the comparison of this quantity with predictions of the high-energy hadronic interaction models QGSJET-II-02, SIBYLL 2.1, QGSJET-II-04 and EPOS-LHC. It reveals that the observed attenuation of the muon content of extensive air showers in the atmosphere is lower than predicted. Differences are, however, less significant with the post-LHC models. It has an effect on the absolute energy and mass composition, but not on the spectral features.

## 3.2 Estimation of muon content

A further analysis is performed by investigating the muon content of air showers as a function of the primary energy in the interval from $10^{16}$ to $10^{18}$ eV and for different zenith angle ranges by means of KASCADE-Grande data [13]. We compared this estimation with the predictions of the muon content for the post-LHC hadronic interaction models including the new model of SIBYLL 2.3c.

For this study, we used the method proposed by the SUGAR [14] experiment to get $N_\mu(E)$ due to the lack of a model independent energy estimator in KASCADE-Grande. The concept of the method is to compare the experimental $N_\mu$ histogram with the corresponding predictions by a reference cosmic ray model based on the spectrum [15] from the Pierre Auger Observatory and relative cosmic-rays abundances from the Global Spline Fit (GSF) [16] model, which is shown in Fig. 5 (left). By a $\chi^2$ minimization procedure, we find a shift between Monte Carlo and measured data that allows to describe the experimental $N_\mu$ distribution:

$$\delta_\mu = \Delta\log_{10}(N_\mu) = a_0 + a_1 \cdot \log_{10}\left(\frac{E}{\text{GeV}}\right) + a_2 \cdot \log_{10}^2\left(\frac{E}{\text{GeV}}\right), \tag{2}$$

where $a_{0,1,2}$ are the fitting parameters. As an example, the result of the fit of the measured data for vertical events is shown in Fig. 5 (right). Applying the fitted shift to the true $N_\mu$ of the Monte Carlo simulations [13], we obtained the actual muon content:

$$\log_{10}[N_\mu(E)] = \log_{10}[N_{\mu,MC}(E)] + \delta_\mu. \tag{3}$$

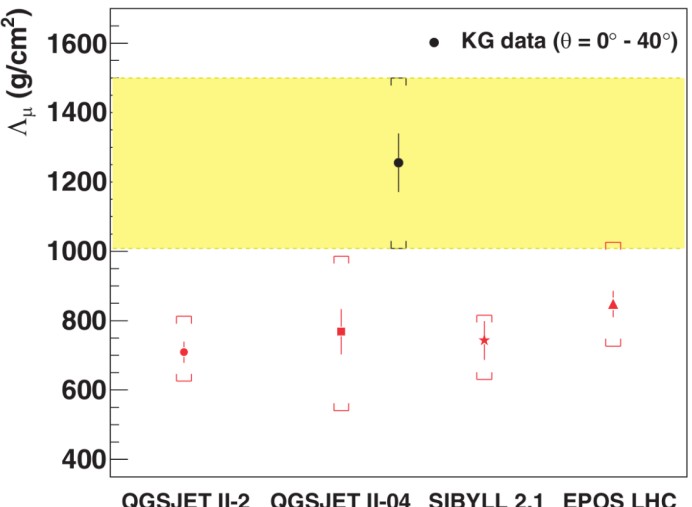

Figure 4: Muon attenuation length measured with the KASCADE-Grande experiment (black) compared with predictions (red) of QGSJET-II-02 and post-LHC high-energy hadronic interaction models [12]. The total uncertainty (statistical and systematic errors added in quadrature) are shown with squared brackets. Error bars represent only statistical uncertainties. The shadowed band covers the total uncertainty estimated for the experimental result.

Figure 6 presents the estimated $\log_{10}[N_\mu/E(\text{GeV})]$ from the measured data as a function of the primary energy of $\log_{10}(E/\text{GeV})$ for each zenith angle interval using fits with different hadronic interaction models. The results with their respective statistical and systematic uncertainties are compared with the fit to the predictions of the post-LHC hadronic models for H (blue), Fe (red), and the reference composition model (dashed line). The dominant contribution to the total systematic errors are uncertainties in the energy calibration and the Lateral Distribution Function (LDF) of muons.

We observed that none of the high-energy interaction models studied here is able to describe consistently the KASCADE-Grande data on $N_\mu$ for all zenith angles and energies. In addition, predictions of EPOS-LHC, SIBYLL 2.3 and SIBYLL 2.3c on $N_\mu$ for primary energies between 100 PeV and 1 EeV are above the KASCADE-Grande data for vertical extensive air showers. However, it shows a better agreement for inclined showers close to 40°. It is also important to note that the observed anomalies could imply that the energy spectrum of muons from real extensive air showers at production site for a given primary energy is harder than the respective model predictions. In addition, studies of the muon content of extensive air showers with the new hadronic model SIBYLL 2.3d are under investigation.

## 4 KASCADE Cosmic-ray Data Centre (KCDC)

KCDC [17] is a web portal (https://kcdc.ikp.kit.edu), where scientific data of the completed KASCADE and KASCADE-Grande experiments are made available for the interested general public. Since the first release in 2013, KCDC provides to the public users the measured and reconstructed parameters of air showers. Over the last years, we have continuously updated the data center with increasing the number of detector components and the data sets from KASCADE-Grande. Full air-shower simulations with the inclusion of the detector responses were published as well. In the latest release, an independent data shop for a specific KASCADE-

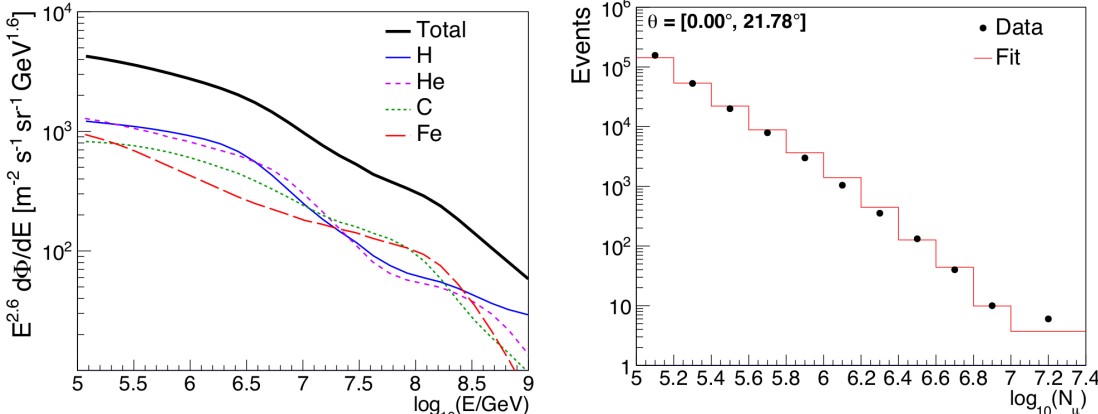

Figure 5: Left: Cosmic ray energy spectra for the mass groups of H, He, C, and Fe in the reference composition model, which is based on the GSF composition model [16] and the Pierre Auger total energy spectrum [15]. Right: Distribution of the measured number of muons $N_\mu$ (black dots) and the result of the fit (red lines) using QGSJET-II-04. Vertical events ($\theta < 21.78°$) are shown here as an example [13].

Grande event selection and data from other air-shower experiments e.g. MAKET-ANI in Atmenia, were included. Moreover, the data points of more than 100 energy spectra from many different experiments were published. A public server with access to a Jupyter Notebook was attached, recently.

For the future, the publication of the accompanying software tools for open access will be achieved. In the next release, a complete upgrade of all software components is scheduled, which is mainly related with the file download server [18]. Moreover, the shared cluster will be completely replaced to speed up the processing of user requests.

## 5 Conclusion

By means of the shower size measured by KASCADE-Grande, the energy spectra of different mass groups were reconstructed based on the new hadronic interaction model SIBYLL 2.3d. It was compared with the spectra based on the different post-LHC interaction models. All structures of the energy spectra confirmed by previous measurements are shown: observation of a heavy knee at around $10^{17}$ eV accompanied by a flattening of the light component at the same energy. This might be a sign of an extra-galactic component and it is already dominant below the energy of $10^{17}$ eV for the case of SIBYLL 2.3d model.

The new model of SIBYLL 2.3d predicts a higher number of muons compared to other hadronic interaction models. Related to this, it predicts the lowest flux of heavy primaries of all models, i.e. the lightest composition. Detailed studies including estimation of systematic uncertainties and the correction of shower fluctuations are in progress.

In addition, we estimated the muon content of extensive air showers as a function of the primary energy using the KASCADE-Grande data in the energy range from $10^{16}$ to $10^{18}$ eV and zenith angle smaller than 40°. The comparison to the predictions of the interaction models reveals discrepancies between the muon data and the models. This might imply that the muon energy spectrum from measured extensive air showers is harder than that from predicted ones at a given primary energy.

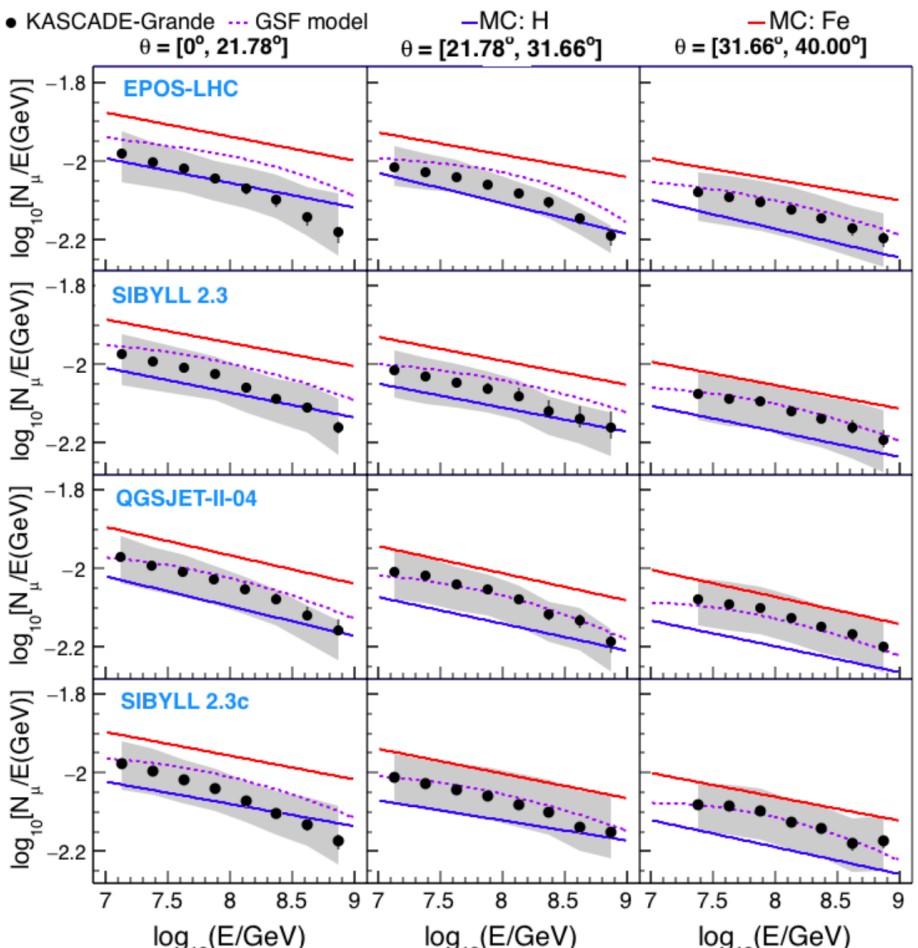

Figure 6: Measured (black dots) and predicted (lines) mean values of $\log_{10}[N_\mu/E(\text{GeV})]$ for different hadronic interaction models of EPOS-LHC, SIBYLL 2.3, QGSJET-II-04, and SIBYLL 2.3c [13]. Each column corresponds to a different zenith angular bin. The red and blue lines represent the predictions for H and Fe, respectively. The purple dashed line is the prediction from the GSF model. The error bars are systematic errors and the gray bands represent the total systematic error.

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
