# Peer review of "Results from recent analysis of KASCADE-Grande data"

_SciPost Physics Proceedings, doi:SciPost Phys. Proc. 13, 036 (2023)_

## Round 2 · Referee Report · Anonymous (Referee 1) · 2022-12-27

Report

The paper describes the KASCADE-GRANDE data analysis of cosmic ray spectrum by classifying showers as electron-poor and rich to represent heavy and light primary groups. It compares with both pre and post-LHC hadronic interaction models. The muon content in the data was also compared with different post-LHC hadronic models. The prediction of the models on the muons have significant discrepancy with the data.

The paper is reasonably well written and easy to follow. Although the the abstract says that the paper investigates the new hadronic interaction model Sibyll 2.3d, but in figure 6, I didn't find any comparison of this model. At least some explanation should have been there why it is not given.

Nevertheless, there is an adequate content for the paper to be published in this journal after taking care of the suggested changes.

Requested changes

  1. Page 3, last para: A remarkable feature is that the concave structure at about 10^16 eV …. Since the first data point starts at about 10^16 eV, it is not clear which feature the authors are referring to?

  2. Page 4, para 2: As far I know, SIBYLL 2.1 is a pre-LHC model not post-LHC. Please clarify.

  3. The authors mention that the all the features of earlier analysis are confirmed. In fig 3, I would be good if the earlier results of KASCADE-GRANDE are plotted here.

  4. Pag 5, Muon content: several experiments exhibited discrepancy…….Please provide a few references for muon puzzle problem.

  5. Page 5, Muon attenuation length: How the muon attenuation length was obtained in simulation for different hadronic interaction models has not been described. At least some references should be provided.

  6. In Figure 4, no comparison with Sibyll 2.3 or Sibyll 2.3c or Sibyll 2.3d? Please clarify.

  7. In Fig 6: No Sibyll 2.3d comparison in this plot? Please clarify.

  • validity: good
  • significance: good
  • originality: good
  • clarity: good
  • formatting: good
  • grammar: good

Author:  Donghwa Kang  on 2023-01-10  [id 3224]

(in reply to Report 1 on 2022-12-27)

Dear Reviewer:

We appreciate your review and comments. We have updated the revised version on the arXiv.

With respect to the comment on the new interaction model of Sibyll 2.3d, we investigated the new model Sibyll 2.3d only with the energy spectra of heavy and light mass groups. Since studies of the muon content with Sibyll 2.3d is still ongoing, the results of muon studies in hand is presented in this paper. We modified the abstract to clarify it.

Please find below our response to each of your comments.

Requested changes 1. Page 3, last para: A remarkable feature is that the concave structure at about 10^16 eV …. Since the first data point starts at about 10^16 eV, it is not clear which feature the authors are referring to? => The concave behavior just above 10^16 eV (the first two points in the spectrum) is not described by a single power law. This hardening of the spectrum is validated by several cross-checks. In addition, a hardening of the spectrum is expected when a pure rigidity dependence of the galactic cosmic rays is assumed. The gap in the knee positions of light primaries and the heavy group can lead to a hardening of the spectrum. A detailed discussion on this structure can be found in Ref. [W.D. Apel et al., KASCADE-Grande collaboration, Astrop. Phys. 36 (2012) 183]. We added this reference to the manuscript.

  1. Page 4, para 2: As far I know, SIBYLL 2.1 is a pre-LHC model not post-LHC. Please clarify. => We clarified.

  2. The authors mention that the all the features of earlier analysis are confirmed. In fig 3, I would be good if the earlier results of KASCADE-GRANDE are plotted here. => In Fig. 3, the earlier results are already plotted, which are the spectra based on QGSjet-II-04, EPOS-LHC, SIBYLL 2.1, and SIBYLL 2.3 models.

  3. Pag 5, Muon content: several experiments exhibited discrepancy…….Please provide a few references for muon puzzle problem. => We provided the reference.

  4. Page 5, Muon attenuation length: How the muon attenuation length was obtained in simulation for different hadronic interaction models has not been described. At least some references should be provided. => We added the reference.

  5. In Figure 4, no comparison with Sibyll 2.3 or Sibyll 2.3c or Sibyll 2.3d? Please clarify. => This result is an earlier work. At that time, the versions Sibyll 2.3, 2.3c, and 2.3d of the interaction model were not available. An update of this result is under consideration.

  6. In Fig 6: No Sibyll 2.3d comparison in this plot? Please clarify. => As mentioned above, studies of the muon content with the new interaction model Sibyll 2.3d is under investigation. In Page 7, we added the sentence “In addition, studies of the muon content of extensive air showers with the new hadronic model SIBYLL 2.3d are under investigation”.

Attachment:

ISVHECRI22_Proc_DKang_Marked_Revision.pdf

---

## Round 3 · Referee Report · Anonymous (Referee 1) · 2023-1-23

Report

Authors have taken care my comments and questions. I now recommend its publication in SciPost.

---

## Editorial Decision

published